# Transition Metal-Free Synthesis of 3-Acylquinolines through Formal [4+2] Annulation of Anthranils and Enaminones

Kai-Ling Zhang [1,†], Jia-Cheng Yang [1,†], Qin Guo [1] and Liang-Hua Zou [1,2,*]

1 Key Laboratory of Carbohydrate Chemistry and Biotechnology, Ministry of Education, School of Life Science and Health Engineering, Jiangnan University, Wuxi 214122, China
2 School of Chemistry and Chemical Engineering, Henan Normal University, Xinxiang 453007, China
* Correspondence: zoulianghua@jiangnan.edu.cn
† These authors contributed equally to this work.

**Abstract:** A transition metal-free protocol has been developed for the synthesis of 3-acyl quinolines through aza-Michael addition and intramolecular annulation of enaminones with anthranils. Both methanesulfonic acid (MSA) and NaI play an important role in the reaction. This ring-opening/reconstruction strategy features easy operation, high yields, broad substrate scope and excellent efficiency.

**Keywords:** quinolines; annulation; transition metal-free; anthranils; enaminones





## 1. Introduction

The quinoline skeleton, as one of the most prevalent heterocycles, widely exist in natural products and synthetic compounds [1–4]. In the past decades, significant attention has been devoted to the synthesis of quinolines because of their valuable biological and chemical properties in the application of medicals [5–9], ligands [10,11], and materials [12–15]. Therefore, the development of novel, practical, and efficient synthetic methods is highly desirable for the synthesis of quinoline derivatives [16,17].

Traditional strategies for the synthesis of quinolines, such as Friedlander synthesis [18] and Doebner–von Miller synthesis [19], could date back to a long time ago. The establishment of quinoline reservoirs nowadays has been broadly prepared by using aromatic amines, isoxazoles, or amino acids as nitrogen sources [20–23]. Transition metal catalysts were normally required for such multi-component coupling and cascade cyclization reactions [24–32]. Most of these methods suffered from some drawbacks such as a complex reaction system, harsh reaction conditions, and complex by-products. Consequently, the development of a transition metal-free protocol for the synthesis of quinoline derivatives appears desirable and synthetically attractive [33–39].

In particular, the synthesis of 3-acylquinolines has attracted significant attention due to their valuable application. Representative methods for the preparation of such compounds include (i) Fe powder-catalyzed reaction of 2-nitrobenzaldehyde with enaminones [40]; (ii) copper-catalyzed one-pot domino reactions starting from 2-aminobenzylalcohols and propiophenones [41]; and (iii) copper-catalyzed annulation of anthranils and saturated ketones [42]. Besides, several transition metal-free protocols have also been developed. In recent years, the application of enaminones [43–45] and anthranils [46–53] as useful building blocks have attracted significant attention in transition metal-catalyzed organic synthesis and construction of a variety of frameworks. Based on our previous work involving sulfoxonium ylides, aldehydes, and anthranils in constructing quinoline derivatives [54–56], we were intrigued to investigate the reactivity of enaminones in the annulation reactions for the construction of heterocycles [57,58]. Herein, we report a transition metal-free protocol for the synthesis of 3-acylquinolines under simple and easy-to-operate conditions.

## 2. Results

The initial screening and optimization of the reaction conditions were conducted with compounds **1a** and **2a** as substrates. Using **1a** and **2a** in a 2:1 ratio (on a 0.2 mmol scale), EtOH (2 mL) as a solvent and methanesulfonic acid (MSA) (0.3 mmol) as an additive, product **3aa** was obtained in 31% yield by heating at 110 °C (Table 1, entry 1). Next, the use of other solvents, such as HFIP, DME, *n*-octanol, dioxane, THF, and DMSO, was investigated but no better results were achieved (Table 1, entries 2–7). No product was obtained in the absence of MSA (Table 1, entry 8). An attempt to employ a mixture of additives MSA and KBr (each 0.3 mmol) in EtOH or THF led to higher yields of 58% and 37%, respectively, indicating that EtOH was more suitable for this reaction (Table 1, entries 9 and 10). Next, we tried to use other salts instead of KBr as an additive, such as KI and NaI, affording product **3aa** in 89% and 90% yields, respectively (Table 1, entries 11 and 12). When the reaction was performed at lower temperatures, the yields dropped significantly, while a higher temperature did not improve the reaction efficiency (Table 1, entries 13–16).

**Table 1.** Optimization of reaction conditions for the synthesis of **3aa** [a].

| Entry | Additive | Solvent | Temp. (°C) | Yield [b] (%) |
|---|---|---|---|---|
| 1 | MSA | EtOH | 110 | 31 |
| 2 | MSA | HFIP | 110 | 5 |
| 3 | MSA | DME | 110 | 11 |
| 4 | MSA | *n*-octanol | 110 | 0 |
| 5 | MSA | *dioxane* | 110 | 18 |
| 6 | MSA | THF | 110 | 29 |
| 7 | MSA | DMSO | 110 | 0 |
| 8 | / | EtOH | 110 | 0 |
| 9 | MSA/KBr | EtOH | 110 | 58 |
| 10 | MSA/KBr | THF | 110 | 37 |
| 11 | MSA/KI | EtOH | 110 | 89 |
| 12 | MSA/NaI | EtOH | 110 | 90 |
| 13 | MSA/NaI | EtOH | 70 | 43 |
| 14 | MSA/NaI | EtOH | 80 | 65 |
| 15 | MSA/NaI | EtOH | 90 | 84 |
| 16 | MSA/NaI | EtOH | 120 | 90 |

[a] Reaction conditions: **1a** (0.4 mmol), **2a** (0.2 mmol), additive (0.3 mmol), EtOH (2 mL), 110 °C, air, 12 h. [b] Isolated yield.

With the optimized reaction conditions in hand (Table 1, entry 10), the scope of the reaction of anthranil **1a** with a variety of enaminones **2** was investigated and the results are summarized in Scheme 1. Substrates with various electron-donating or electron-withdrawing substituents were applied to the reaction system to provide a series of 3-acylquinolines. For example, the reaction of substrates bearing electron-donating substituents such as -Me, -OMe, -OPh, -NMe$_2$, and -SMe afforded the corresponding products **3ab**–**3af** in good yields ranging from 60% to 84%. For the substrates with -F, -Cl, -Br, and -I groups at the *para*-position, the reaction also proceeded well to give the corresponding products **3ag**–**3ai** in high yields. Substrates containing substituents -CN, -NO$_2$, and -CF$_3$ were also successfully employed in the reaction, yielding products **3ak**–**3am** in moderate yields. The reaction of one substrate bearing a phenyl group afforded the corresponding product **3an** in 81% yield. It appeared that the steric factors did not influence the efficiency significantly. For example, various substituents at the ortho- or meta-position were well tolerated, providing the corresponding products **3ao**–**3at** in similarly good yields ranging from 81% to 88%.

Substrates bearing two substituents at the ortho- and meta-positions were also investigated, affording products **3au–3aw** in 83%, 82%, and 81% yields, respectively. Furthermore, one substrate bearing electron-donating dioxolo-group was also successfully employed in the reaction, providing the desired product **3ax** in 79% yield. The reaction of the substrate bearing naphthyl group afforded product **3ay** in 84% yield. At last, substrates containing oxygen- or sulfur-heterocycle also reacted well with **1a**, giving products **3az** and **3az'** in 83% and 80% yields, respectively.

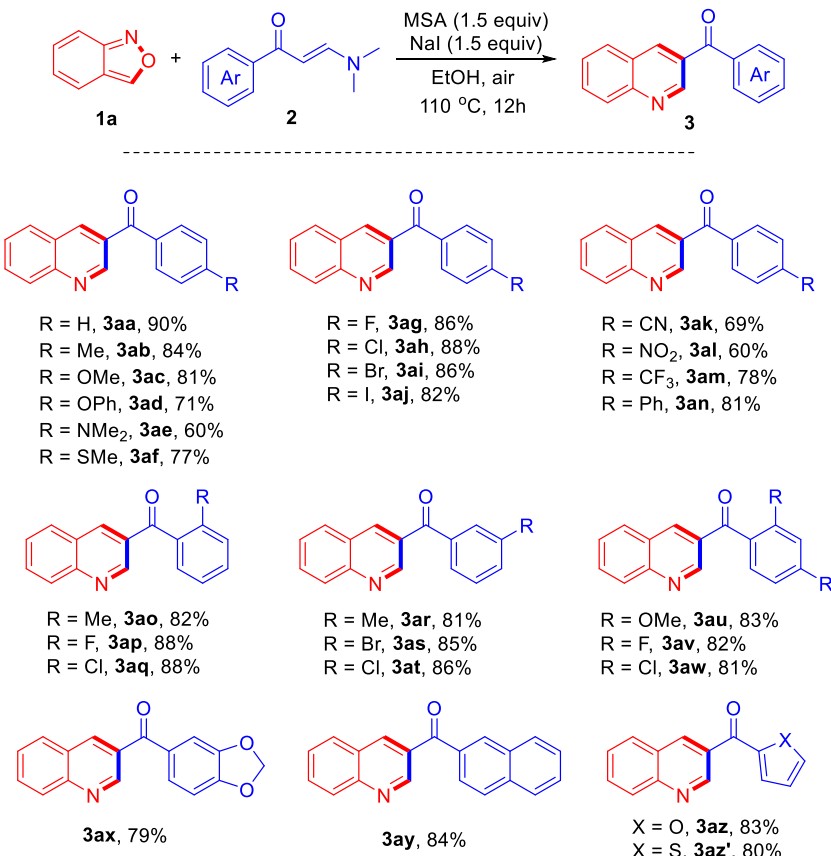

R = H, **3aa**, 90%
R = Me, **3ab**, 84%
R = OMe, **3ac**, 81%
R = OPh, **3ad**, 71%
R = NMe$_2$, **3ae**, 60%
R = SMe, **3af**, 77%

R = F, **3ag**, 86%
R = Cl, **3ah**, 88%
R = Br, **3ai**, 86%
R = I, **3aj**, 82%

R = CN, **3ak**, 69%
R = NO$_2$, **3al**, 60%
R = CF$_3$, **3am**, 78%
R = Ph, **3an**, 81%

R = Me, **3ao**, 82%
R = F, **3ap**, 88%
R = Cl, **3aq**, 88%

R = Me, **3ar**, 81%
R = Br, **3as**, 85%
R = Cl, **3at**, 86%

R = OMe, **3au**, 83%
R = F, **3av**, 82%
R = Cl, **3aw**, 81%

**3ax**, 79%

**3ay**, 84%

X = O, **3az**, 83%
X = S, **3az'**, 80%

**Scheme 1.** Reaction conditions: **1a** (0.4 mmol), **2** (0.2 mmol), methanesulfonic acid (MSA) (0.3 mmol), NaI (0.3 mmol), EtOH (2 mL), 110 °C, air, 12 h.

In order to further explore the scope of the reaction, a series of anthranil **1** were introduced to react with compound **2a** under the optimized reaction conditions, and the results are summarized in Scheme 2. The reactions of various anthranil-bearing electron-donating substituent -OMe proceeded well, affording product **3ba** in 83% yield. The substrate bearing halogen substituents -F, -Cl, and -Br could also be employed in the reaction, providing the corresponding products **3ca–3ea** in high yields. Under the conditions, products **3fa–3ha** with halogen substituents -F, -Cl, and -Br at 7-position were also successfully obtained in yields ranging from 83% to 88%. The reactions of anthranils bearing electron-withdrawing substituents -CF$_3$ and -CO$_2$Me provided products **3ia** and **3ja** in 80% and 75% yields, respectively. In addition, bisubstitued substrates were also investigated, affording the corresponding products **3ka–3ma** in satisfying yields. Finally, one C3-aryl substituted example was investigated in the reaction, giving product **3na** in 84% yield.

**Scheme 2.** Reaction conditions: **1** (0.4 mmol), **2a** (0.2 mmol), methanesulfonic acid (MSA) (0.3 mmol), NaI (0.3 mmol), EtOH (2 mL), 110 °C, air, 12 h.

To test the scalability of the new protocol, gram-scale preparation was carried out under the standard conditions with 12 mmol of **1a** and 6 mmol of **2a**, and product **3aa** was successfully obtained in 85% yield (1.192 g) (Scheme 3).

**Scheme 3.** Gram-scale reaction.

To elucidate the possible mechanism, several preliminary experiments were carried out. At first, 2-aminobenzaldehyde was employed instead of anthranil **2a** to investigate whether it was an intermediate in the process, affording product **3aa** in 56% yield (Scheme 4, Equation (a)). The results showed that 2-aminobenzaldehyde might be an intermediate for the transformation of anthranil **2a**. Furthermore, radical scavengers, such as 2,2,6,6-tetramethyl-1-piperidinyloxy (TEMPO), 2,6-bis(1,1-dimethylethyl)-4-methylphenol (BHT), and 1,1-diphenyltheylene, were tested in the reaction, affording the desired product **3aa** in 70%, 87%, and 81% yields, respectively, indicating that there might be no radical step in the process (Scheme 4, Equation (b)). The [1]H NMR spectrum of the intermolecular competition experiment between (*E*)-3-(dimethylamino)-1-(p-tolyl)prop-2-en-1-one (**2b**) and (*E*)-3-(dimethylamino)-1-(4-fluorophenyl)prop-2-en-1-one (**2g**) showed that the ratio of products **3ab** and **3ag** was 1:0.64. When the electron-deficient substrate was changed to (*E*)-3-(dimethylamino)-1-(4-(trifluoromethyl)phenyl)prop-2-en-1-one (**2m**), the ratio of products **3ab** and **3am** in the intermolecular competition experiment was 1:0.62. Both

results indicated that substrates bearing electron-donating substituents were more active than that with electron-withdrawing substituents (Scheme 4, Equation (c)).

**Scheme 4.** Control reactions.

On the basis of our results and previous reports on enaminone **2a** [59], a possible mechanism on the synthesis of **3aa** is proposed as shown in Scheme 5. First, **2a** is activated by methanesulfonic acid (MSA) to form intermediate **A**, which is attacked by **1a** to give intermediate **B**. NaI is crucial for the success of this reaction, probably due to the fact that it can promote the stability of intermediate **B**. Then, intermediate **C** is produced, affording **D** through an intramolecular nucleophilic addition. Finally, **D** is converted into the desired product **3aa**.

**Scheme 5.** Proposed mechanism for the synthesis of **3aa**.

## 3. Conclusions

In conclusion, a mild and efficient protocol has been developed for the synthesis of a broad range of 3-acyl quinolines under transition metal-free conditions. Various enaminones and anthranils were well employed in the reaction, providing a series of 3-acyl quinolines through aza-Michael addition and intramolecular annulation. Mechanism

studies showed that no radical step was involved in the process. It was noteworthy that MSA and NaI played a significant role for the success of the transformation.

## 4. Experimental Section

General Information. All solvents were dried over molecular sieves. Unless otherwise noted, materials obtained from commercial suppliers were used without further purification. Anthranil **1n** was purchased and other anthranils were all synthesized according to our previous work [55]. All the substrates enaminones were prepared according to the reference [60]. The products were isolated using column chromatography on silica gel (200–300 mesh) by using petroleum ether (PE, 30–60 °C) and ethyl acetate (EA) as eluents. All yields described herein were the isolated yields after column chromatography. Reaction progress and product mixtures were routinely monitored by using TLC using TLC SiO$_2$ sheets, and compounds were visualized under ultraviolet light. $^1$H NMR, $^{13}$C NMR, and $^{19}$F NMR spectra were recorded on a Bruker AVANCE III 400 spectrometer. The spectra were recorded using CDCl$_3$ as a solvent. $^1$H NMR chemical shifts are referenced to tetramethylsilane (TMS, 0 ppm). $^{19}$F NMR chemical shifts are referenced to *p*-fluorotoluene (*p*-fluorotoluene, 0 ppm). Abbreviations are as follows: s (singlet), d (doublet), t (triplet), q (quartet), m (multiplet). High-resolution mass spectra (HRMS) were recorded on Agilent 6540 or FTICR-MS bruker7 T. Melting points were measured with a melting point instrument (Shanghai Yidian Physical Optical Instrument Co., Ltd., Shanghai, China, SGW, X-4A) and were uncorrected.

General procedure for the synthesis of **3aa**: A mixture of **1a** (0.4 mmol, 2.0 equiv), **2a** (0.2 mmol, 1.0 equiv) and NaI (0.3 mmol, 1.5 equiv) were added in a Schlenk tube equipped with a stirring bar. Dry EtOH (2 mL) and MSA (0.3 mmol, 1.5 equiv) were added and the mixture was stirred at 110 °C in a pre-heated oil bath for 12 h under air atmosphere. Then, the mixture was cooled to room temperature and concentrated in vacuo, and the resulting residue was purified by using column chromatography on silica gel with EtOAc/petroleum ether, affording product **3aa** as a white solid in 90% yield (42.1 mg).

### 4.1. Phenyl(quinolin-3-yl)methanone (3aa)

Compound **3aa** was synthesized in accordance with the typical procedure. Purification using column chromatography on silica gel (PE:EA = 10:1) afforded product **3aa** (42.1 mg, 90%) as a white solid.
$^1$H NMR (400 MHz, CDCl$_3$) $\delta$ ppm: 9.31 (d, $J$ = 2.3 Hz, 1H), 8.54 (d, $J$ = 2.2 Hz, 1H), 8.18 (d, $J$ = 8.5 Hz, 1H), 7.91 (d, $J$ = 8.3 Hz, 1H), 7.84 (td, $J$ = 8.2, 2.1 Hz, 3H), 7.63 (q, $J$ = 7.3 Hz, 2H), 7.53 (t, $J$ = 7.7 Hz, 2H). $^{13}$C{$^1$H} NMR (101 MHz, CDCl$_3$) $\delta$ ppm: 194.8, 150.3, 149.4, 138.8, 137.0, 133.0, 131.8, 130.04, 130.0, 129.4, 129.1, 128.6, 127.6, 126.6. HRMS *m/z* (ESI) calcd for C$_{16}$H$_{12}$NO (M+H)$^+$ 234.0913, found 234.0920. The NMR spectra are consistent with the reported literature [31].

### 4.2. Quinolin-3-yl(p-tolyl)methanone (3ab)

Compound **3ab** was synthesized in accordance with the typical procedure. Purification using column chromatography on silica gel (PE:EA = 10:1) afforded product **3ab** (41.4 mg, 84%) as a white solid. $^1$H NMR (400 MHz, CDCl$_3$) $\delta$ ppm: 9.30 (d, $J$ = 2.1 Hz, 1H), 8.54 (d, $J$ = 2.1 Hz, 1H), 8.19 (d, $J$ = 8.4 Hz, 1H), 7.91 (d, $J$ = 9.5 Hz, 1H), 7.84 (t, $J$ = 7.7 Hz, 1H), 7.78 (d, $J$ = 8.1 Hz, 2H), 7.63 (t, $J$ = 7.0 Hz, 1H), 7.33 (d, $J$ = 7.9 Hz, 2H), 2.47 (s, 3H). $^{13}$C{$^1$H} NMR (101 MHz, CDCl$_3$) $\delta$ ppm: 194.5, 150.3, 149.3, 144.0, 138.6, 134.3, 131.7, 130.4, 130.2, 129.4, 129.3, 129.1, 127.5, 126.6, 21.7. The NMR spectra are consistent with the reported literature [31].

### 4.3. (4-Methoxyphenyl)(quinolin-3-yl)methanone (3ac)

Compound **3ac** was synthesized in accordance with the typical procedure. Purification using column chromatography on silica gel (PE:EA = 10:1) afforded product **3ac** (42.6 mg, 81%) as a white solid.

$^1$H NMR (400 MHz, CDCl$_3$) $\delta$ ppm: 9.26 (d, *J* = 2.2 Hz, 1H), 8.50 (d, *J* = 2.2 Hz, 1H), 8.18 (d, *J* = 8.5 Hz, 1H), 7.87 (td, *J* = 13.7, 8.6 Hz, 4H), 7.65–7.59 (m, 1H), 7.00 (d, *J* = 8.8 Hz, 2H), 3.89 (s, 3H). $^{13}$C{$^1$H} NMR (101 MHz, CDCl$_3$) $\delta$ ppm: 193.4, 163.7, 150.2, 149.2, 138.1, 132.5, 131.5, 130.8, 129.7, 129.4, 129.0, 127.4, 126.6, 113.9, 55.5. The NMR spectra are consistent with the reported literature [31].

### 4.4. (4-Phenoxyphenyl)(quinolin-3-yl)methanone (**3ad**)

Compound **3ad** was synthesized in accordance with the typical procedure. Purification using column chromatography on silica gel (PE:EA = 10:1) afforded product **3ad** (46.0 mg, 71%) as a white solid, m.p. 78–80 °C; $^1$H NMR (400 MHz, CDCl$_3$) $\delta$ ppm: 9.29 (d, *J* = 2.1 Hz, 1H), 8.53 (d, *J* = 2.2 Hz, 1H), 8.18 (d, *J* = 8.5 Hz, 1H), 7.93–7.87 (m, 2H), 7.87–7.81 (m, 2H), 7.63 (t, *J* = 7.5 Hz, 1H), 7.41 (t, *J* = 7.9 Hz, 2H), 7.21 (t, *J* = 7.4 Hz, 1H), 7.09 (dd, *J* = 18.3, 8.7 Hz, 4H). $^{13}$C{$^1$H} NMR (101 MHz, CDCl$_3$) $\delta$ ppm: 193.4, 162.2, 155.2, 150.2, 149.3, 138.3, 132.4, 131.6, 131.3, 130.4, 130.1, 129.4, 129.0, 127.5, 126.6, 124.8, 120.3, 117.3. HRMS *m/z* (ESI) calcd for C$_{22}$H$_{15}$NO$_2$(M+H)$^+$ 326.1176, found 326.1183.

### 4.5. (4-(Dimethylamino)phenyl)(quinolin-3-yl)methanone (**3ae**)

Compound **3ae** was synthesized in accordance with the typical procedure. Purification using column chromatography on silica gel (PE:EA = 5:1) afforded product **3ae** (33.1 mg, 60%) as a yellow solid, m.p. 118–120 °C; $^1$H NMR (400 MHz, CDCl$_3$) $\delta$ ppm: 9.25 (d, *J* = 2.2 Hz, 1H), 8.49 (d, *J* = 3.1 Hz, 1H), 8.18 (d, *J* = 8.5 Hz, 1H), 7.91 (d, *J* = 7.9 Hz, 1H), 7.85–7.79 (m, 3H), 7.62 (ddd, *J* = 8.1, 6.9, 1.2 Hz, 1H), 6.73–6.69 (m, 2H), 3.10 (s, 6H). $^{13}$C{$^1$H} NMR (101 MHz, CDCl$_3$) $\delta$ ppm: 192.7, 153.6, 150.4, 148.9, 137.6, 132.7, 131.8, 131.1, 129.3, 128.9, 127.3, 126.8, 124.2, 110.7, 40.0. HRMS *m/z* (ESI) calcd for C$_{18}$H$_{16}$N$_2$O (M+H)$^+$ 277.1335, found 277.1327.

### 4.6. (Methylthio)phenyl)(quinolin-3-yl)methanone (**3af**)

Compound **3af** was synthesized in accordance with the typical procedure. Purification using column chromatography on silica gel (PE:EA = 10:1) afforded product **3af** (43.1 mg, 77%) as a white solid, m.p. 110–112 °C; $^1$H NMR (400 MHz, CDCl$_3$) $\delta$ ppm: 9.27 (d, *J* = 2.1 Hz, 1H), 8.50 (d, *J* = 2.2 Hz, 1H), 8.17 (d, *J* = 8.5 Hz, 1H), 7.89 (d, *J* = 8.2 Hz, 1H), 7.84–7.76 (m, 3H), 7.61 (t, *J* = 7.5 Hz, 1H), 7.31 (d, *J* = 8.2 Hz, 2H), 2.53 (s, 3H). $^{13}$C{$^1$H} NMR (101 MHz, CDCl$_3$) $\delta$ ppm: 193.7, 150.2, 149.3, 146.3, 138.3, 133.0, 131.6, 130.5, 130.3, 129.4, 129.0, 127.5, 126.5, 124.9, 14.7. HRMS *m/z* (ESI) calcd for C$_{17}$H$_{13}$NOS (M+H)$^+$ 280.0791, found 280.0784.

### 4.7. (4-Fluorophenyl)(quinolin-3-yl)methanone (**3ag**)

Compound **3ag** was synthesized in accordance with the typical procedure. Purification using column chromatography on silica gel (PE:EA = 10:1) afforded product **3ag** (43.0 mg, 86%) as a white solid.

$^1$H NMR (400 MHz, CDCl$_3$) $\delta$ ppm: 9.28 (d, *J* = 2.2 Hz, 1H), 8.52 (d, *J* = 2.8 Hz, 1H), 8.18 (d, *J* = 8.6 Hz, 1H), 7.93–7.88 (m, 3H), 7.85 (ddd, *J* = 8.5, 6.9, 1.5 Hz, 1H), 7.66–7.62 (m, 1H), 7.21 (t, *J* = 8.6 Hz, 2H). $^{13}$C{$^1$H} NMR (101 MHz, CDCl$_3$) $\delta$ ppm: 193.4, 165.7 (d, *J* = 255.4 Hz), 150.1, 149.4, 138.6, 133.2 (d, *J* = 2.8 Hz), 132.6 (d, *J* = 9.4 Hz), 131.9, 129.3, 127.7, 126.5, 115.9 (d, *J* = 21.9 Hz). $^{19}$F NMR (376 MHz, CDCl$_3$) $\delta$ ppm: −104.4. The NMR spectra are consistent with the reported literature [31].

### 4.8. (4-Chlorophenyl)(quinolin-3-yl)methanone (**3ah**)

Compound **3ah** was synthesized in accordance with the typical procedure. Purification using column chromatography on silica gel (PE:EA = 10:1) afforded product **3ah** (47.1 mg, 88%) as a white solid. $^1$H NMR (400 MHz, CDCl$_3$) $\delta$ ppm: 9.28 (d, *J* = 2.2 Hz, 1H), 8.52 (d, *J* = 2.3 Hz, 1H), 8.18 (d, *J* = 8.5 Hz, 1H), 7.92–7.83 (m, 2H), 7.82–7.79 (m, 2H), 7.64 (ddd, *J* = 8.1, 6.9, 1.2 Hz, 1H), 7.53–7.46 (m, 2H). $^{13}$C{$^1$H} NMR (101 MHz, CDCl$_3$) $\delta$ ppm: 193.6,

150.0, 149.5, 139.6, 138.7, 135.2, 132.0, 131.4, 129.6, 129.4, 129.1, 129.0, 127.7, 126.5. The NMR spectra are consistent with the reported literature [31].

### 4.9. (4-Bromophenyl)(quinolin-3-yl)methanone (3ai)

Compound **3ai** was synthesized in accordance with the typical procedure. Purification using column chromatography on silica gel (PE:EA = 10:1) afforded product **3ai** (53.5 mg, 86%) as a white solid. $^1$H NMR (400 MHz, CDCl$_3$) $\delta$ ppm: 9.29 (d, *J* = 2.2 Hz, 1H), 8.53 (d, *J* = 1.3 Hz, 1H), 8.19 (d, *J* = 9.5 Hz, 1H), 7.92 (d, *J* = 8.1 Hz, 1H), 7.86 (ddd, *J* = 8.5, 6.9, 1.4 Hz, 1H), 7.76–7.72 (m, 2H), 7.70–7.63 (m, 3H). $^{13}$C{$^1$H} NMR (101 MHz, CDCl$_3$) $\delta$ ppm: 193.8, 150.1, 149.5, 138.8, 135.7, 132.0, 132.0, 131.5, 129.6, 129.5, 129.1, 128.3, 127.7, 126.5. The NMR spectra are consistent with the reported literature [31].

### 4.10. (4-Iodophenyl)(quinolin-3-yl)methanone (3aj)

Compound **3aj** was synthesized in accordance with the typical procedure. Purification using column chromatography on silica gel (PE:EA = 10:1) afforded product **3aj** (66.3 mg, 82%) as a white solid. $^1$H NMR (400 MHz, CDCl$_3$) $\delta$ ppm: 9.28 (d, *J* = 2.2 Hz, 1H), 8.51 (d, *J* = 1.5 Hz, 1H), 8.18 (d, *J* = 8.5 Hz, 1H), 7.92–7.83 (m, 4H), 7.66–7.60 (m, 1H), 7.59–7.53 (m, 2H). $^{13}$C{$^1$H} NMR (101 MHz, CDCl$_3$) $\delta$ ppm: 194.0, 150.0, 149.5, 138.7, 137.9, 136.2, 132.0, 131.3, 129.5, 129.4, 127.7, 126.4, 101.0. The NMR spectra are consistent with the reported literature [22].

### 4.11. 4-(Quinoline-3-carbonyl)benzonitrile (3ak)

Compound **3ak** was synthesized in accordance with the typical procedure. Purification using column chromatography on silica gel (PE:EA = 5:1) afforded product **3ak** (35.5 mg, 69%) as a white solid. $^1$H NMR (400 MHz, CDCl$_3$) $\delta$ ppm: 9.28 (d, *J* = 2.2 Hz, 1H), 8.51 (d, *J* = 2.2 Hz, 1H), 8.18 (d, *J* = 8.4 Hz, 1H), 7.92 (t, *J* = 7.4 Hz, 3H), 7.88 (d, *J* = 7.5 Hz, 1H), 7.83 (d, *J* = 8.1 Hz, 2H), 7.65 (t, *J* = 7.5 Hz, 1H). $^{13}$C{$^1$H} NMR (101 MHz, CDCl$_3$) $\delta$ ppm: 193.3, 149.8, 149.7, 140.4, 139.0, 132.4, 132.4, 130.1, 129.5, 129.2, 128.8, 127.9, 126.4, 117.7, 116.2. The NMR spectra are consistent with the reported literature [31].

### 4.12. (4-Nitrophenyl)(quinolin-3-yl)methanone (3al)

Compound **3al** was synthesized in accordance with the typical procedure. Purification using column chromatography on silica gel (PE:EA = 5:1) afforded product **3al** (35.4 mg, 60%) as a white solid. $^1$H NMR (400 MHz, CDCl$_3$) $\delta$ ppm: 9.32 (d, *J* = 2.2 Hz, 1H), 8.54 (d, *J* = 2.2 Hz, 1H), 8.39 (d, *J* = 8.7 Hz, 2H), 8.20 (d, *J* = 8.5 Hz, 1H), 8.01 (d, *J* = 8.8 Hz, 2H), 7.93 (d, *J* = 8.3 Hz, 1H), 7.89 (t, *J* = 7.6 Hz, 1H), 7.70–7.64 (m, 1H). $^{13}$C{$^1$H} NMR (101 MHz, CDCl$_3$) $\delta$ ppm: 193.1, 150.2, 149.8, 142.1, 139.1, 132.5, 130.7, 129.6, 129.3, 128.8, 128.0, 126.4, 123.8. The NMR spectra are consistent with the reported literature [31].

### 4.13. Quinolin-3-yl(4-(trifluoromethyl)phenyl)methanone (3am)

Compound **3am** was synthesized in accordance with the typical procedure. Purification using column chromatography on silica gel (PE:EA = 10:1) afforded product **3am** (47.6 mg, 78%) as a white solid. $^1$H NMR (400 MHz, CDCl$_3$) $\delta$ ppm: 9.32 (d, *J* = 2.2 Hz, 1H), 8.54 (d, *J* = 2.3 Hz, 1H), 8.20 (d, *J* = 8.5 Hz, 1H), 7.92 (d, *J* = 8.2 Hz, 3H), 7.88 (ddd, *J* = 8.5, 6.9, 1.5 Hz, 1H), 7.81 (d, *J* = 7.9 Hz, 2H), 7.66 (ddd, *J* = 8.1, 6.9, 1.1 Hz, 1H). $^{13}$C{$^1$H} NMR (101 MHz, CDCl$_3$) $\delta$ ppm: 193.8, 150.0, 149.7, 140.0, 139.1, 134.3 (q, *J* = 32.7 Hz), 132.3, 130.1, 129.5, 129.2, 129.1, 127.8, 126.4, 125.7 (q, *J* = 3.5 Hz), 123.5 (q, *J* = 272.5 Hz). $^{19}$F NMR (376 MHz, CDCl$_3$) $\delta$ ppm: −62.9. The NMR spectra are consistent with the reported literature [31].

### 4.14. [1,1′-Biphenyl]-4-yl(quinolin-3-yl)methanone (3an)

Compound **3an** was synthesized in accordance with the typical procedure. Purification using column chromatography on silica gel (PE:EA = 10:1) afforded product **3an** (50.2 mg, 81%) as a white solid.

[1]H NMR (400 MHz, CDCl$_3$) $\delta$ ppm: 9.35 (d, *J* = 2.2 Hz, 1H), 8.59 (d, *J* = 2.1 Hz, 1H), 8.21 (d, *J* = 8.5 Hz, 1H), 7.94 (t, *J* = 8.3 Hz, 3H), 7.85 (t, *J* = 8.4 Hz, 1H), 7.75 (d, *J* = 8.1 Hz, 2H), 7.65 (t, *J* = 6.6 Hz, 3H), 7.45 (dt, *J* = 29.3, 7.4 Hz, 3H). [13]C{[1]H} NMR (101 MHz, CDCl$_3$) $\delta$ ppm: 194.3, 150.2, 149.4, 145.8, 139.6, 138.6, 135.6, 131.7, 130.6, 130.2, 129.4, 129.1, 129.0, 128.3, 127.5, 127.2, 127.2, 126.6. The NMR spectra are consistent with the reported literature [22].

### 4.15. Quinolin-3-yl(o-tolyl)methanone (**3ao**)

Compound **3ao** was synthesized in accordance with the typical procedure. Purification using column chromatography on silica gel (PE:EA = 10:1) afforded product **3ao** (40.4 mg, 82%) as a white solid.

[1]H NMR (400 MHz, CDCl$_3$) $\delta$ ppm: 9.36 (d, *J* = 2.2 Hz, 1H), 8.47 (d, *J* = 2.2 Hz, 1H), 8.18 (d, *J* = 7.4 Hz, 1H), 7.89–7.82 (m, 2H), 7.64–7.59 (m, 1H), 7.46 (t, *J* = 7.5 Hz, 1H), 7.40–7.34 (m, 2H), 7.32–7.28 (m, 1H), 2.40 (s, 3H). [13]C{[1]H} NMR (101 MHz, CDCl$_3$) $\delta$ ppm: 197.0, 150.3, 149.7, 139.4, 137.5, 137.3, 132.1, 131.4, 131.0, 130.1, 129.4, 129.3, 128.9, 127.5, 126.7, 125.4, 20.1. The NMR spectra are consistent with the reported literature [31].

### 4.16. (2-Fluorophenyl)(quinolin-3-yl)methanone (**3ap**)

Compound **3ap** was synthesized in accordance with the typical procedure. Purification using column chromatography on silica gel (PE:EA = 10:1) afforded product **3ap** (44.1 mg, 88%) as a white solid.

[1]H NMR (400 MHz, CDCl$_3$) $\delta$ ppm: 9.33 (s, 1H), 8.54 (s, 1H), 8.17 (d, *J* = 8.5 Hz, 1H), 7.92–7.82 (m, 2H), 7.69–7.58 (m, 3H), 7.33 (td, *J* = 7.5, 1.1 Hz, 1H), 7.21 (t, *J* = 9.2 Hz, 1H). [13]C{[1]H} NMR (101 MHz, CDCl$_3$) $\delta$ ppm: 191.8, 160.2 (d, *J* = 253.0 Hz), 158.9, 149.8, 134.0 (d, *J* = 8.4 Hz), 134.0, 132.2, 131.0 (d, *J* = 2.6 Hz), 129.9, 129.4, 127.5, 126.1, 126.0, 124.7 (d, *J* = 3.4 Hz), 116.5 (d, *J* = 21.8 Hz). [19]F NMR (376 MHz, CDCl$_3$) $\delta$ ppm: −109.7. The NMR spectra are consistent with the reported literature [31].

### 4.17. (2-chlorophenyl)(quinolin-3-yl)methanone (**3aq**)

Compound **3aq** was synthesized in accordance with the typical procedure. Purification using column chromatography on silica gel (PE:EA = 10:1) afforded product **3aq** (47.2 mg, 88%) as a white solid.

[1]H NMR (400 MHz, CDCl$_3$) $\delta$ ppm: 9.34 (d, *J* = 2.2 Hz, 1H), 8.47 (d, *J* = 3.0 Hz, 1H), 8.17 (d, *J* = 8.4 Hz, 1H), 7.90–7.83 (m, 2H), 7.63–7.59 (m, 1H), 7.51 (d, *J* = 3.0 Hz, 2H), 7.49–7.41 (m, 2H). [13]C{[1]H}NMR (101 MHz, CDCl$_3$) $\delta$ ppm: 193.9, 149.9, 139.5, 137.6, 132.4, 131.8, 131.4, 130.3, 129.5, 129.4, 128.9, 127.6, 127.0, 126.7. The NMR spectra are consistent with the reported literature [31].

### 4.18. Quinolin-3-yl(m-tolyl)methanone (**3ar**)

Compound **3ar** was synthesized in accordance with the typical procedure. Purification using column chromatography on silica gel (PE:EA = 10:1) afforded product **3ar** (40.1 mg, 81%) as a white solid.

[1]H NMR (400 MHz, CDCl$_3$) $\delta$ ppm: 9.31 (d, *J* = 2.2 Hz, 1H), 8.55 (d, *J* = 2.2 Hz, 1H), 8.19 (d, *J* = 8.5 Hz, 1H), 7.92 (d, *J* = 8.1 Hz, 1H), 7.85 (ddd, *J* = 8.5, 6.9, 1.5 Hz, 1H), 7.69–7.61 (m, 3H), 7.48–7.39 (m, 2H), 2.44 (s, 3H). [13]C{[1]H} NMR (101 MHz, CDCl$_3$) $\delta$ ppm: 195.1, 150.3, 149.4, 138.8, 138.6, 137.0, 133.8, 131.8, 130.4, 130.2, 129.4, 129.1, 128.4, 127.5, 127.3, 126.6, 21.3. The NMR spectra are consistent with the reported literature [31].

### 4.19. (3-Bromophenyl)(quinolin-3-yl)methanone (**3as**)

Compound **3as** was synthesized in accordance with the typical procedure. Purification using column chromatography on silica gel (PE:EA = 10:1) afforded product **3as** (54.1 mg, 86%) as a white solid.

[1]H NMR (400 MHz, CDCl$_3$) $\delta$ ppm: 9.28 (d, *J* = 2.2 Hz, 1H), 8.53 (d, *J* = 2.2 Hz, 1H), 8.18 (d, *J* = 8.4 Hz, 1H), 7.99 (s, 1H), 7.92 (d, *J* = 9.5 Hz, 1H), 7.87–7.83 (m, 1H), 7.78–7.72 (m, 2H), 7.63 (t, *J* = 7.5 Hz, 1H), 7.40 (t, *J* = 7.8 Hz, 1H). [13]C{[1]H} NMR (101 MHz, CDCl$_3$)

$\delta$ ppm: 193.3, 150.0, 149.6, 138.8, 135.8, 132.7, 132.1, 130.1, 129.5, 129.4, 129.2, 128.4, 127.7, 126.5, 122.9. The NMR spectra are consistent with the reported literature [35].

### 4.20. (3-Chlorophenyl)(quinolin-3-yl)methanone (3at)

Compound **3at** was synthesized in accordance with the typical procedure. Purification using column chromatography on silica gel (PE:EA = 10:1) afforded product **3at** (43.2 mg, 85%) as a white solid.

$^1$H NMR (400 MHz, CDCl$_3$) $\delta$ ppm: 9.29 (d, $J$ = 2.2 Hz, 1H), 8.53 (d, $J$ = 2.2 Hz, 1H), 8.18 (d, $J$ = 8.5 Hz, 1H), 7.92 (d, $J$ = 9.6 Hz, 1H), 7.87–7.82 (m, 2H), 7.70 (d, $J$ = 7.6 Hz, 1H), 7.66–7.59 (m, 2H), 7.46 (t, $J$ = 7.8 Hz, 1H). $^{13}$C{$^1$H} NMR (101 MHz, CDCl$_3$) $\delta$ ppm: 193.4, 150.0, 149.6, 138.8, 138.6, 135.0, 132.9, 132.0, 129.9, 129.8, 129.5, 129.4, 129.2, 128.0, 127.7, 126.5. The NMR spectra are consistent with the reported literature [31].

### 4.21. (2,4-Dimethoxyphenyl)(quinolin-3-yl)methanone (3au)

Compound **3au** was synthesized in accordance with the typical procedure. Purification using column chromatography on silica gel (PE:EA = 10:1) afforded product **3au** (48.6 mg, 83%) as a yellow solid, m.p. 82–84 °C; $^1$H NMR (400 MHz, CDCl$_3$) $\delta$ ppm: 9.18 (d, $J$ = 2.2 Hz, 1H), 8.53 (d, $J$ = 2.2 Hz, 1H), 8.14 (d, $J$ = 9.3 Hz, 1H), 7.90 (d, $J$ = 8.3 Hz, 1H), 7.80 (t, $J$ = 7.7 Hz, 1H), 7.58 (d, $J$ = 8.4 Hz, 2H), 6.62 (dd, $J$ = 8.5, 2.2 Hz, 1H), 6.52 (d, $J$ = 2.2 Hz, 1H), 3.89 (s, 3H), 3.65 (s, 3H). $^{13}$C{$^1$H} NMR (101 MHz, CDCl$_3$) $\delta$ ppm: 193.7, 164.4, 159.9, 150.7, 149.3, 138.1, 133.0, 131.8, 131.6, 129.4, 127.3, 127.1, 120.7, 105.4, 98.8, 55.7, 55.6. HRMS $m/z$ (ESI) calcd for C$_{18}$H$_{15}$NO$_3$ (M+H)$^+$ 294.1125, found 294.1133.

### 4.22. (2,4-Difluorophenyl)(quinolin-3-yl)methanone (3av)

Compound **3av** was synthesized in accordance with the typical procedure. Purification using column chromatography on silica gel (PE:EA = 10:1) afforded product **3av** (44.1 mg, 82%) as a white solid, m.p. 94–96 °C; $^1$H NMR (400 MHz, CDCl$_3$) $\delta$ ppm: 9.29 (s, 1H), 8.52 (s, 1H), 8.17 (d, $J$ = 7.5 Hz, 1H), 7.91 (d, $J$ = 8.2 Hz, 1H), 7.87–7.82 (m, 1H), 7.76–7.69 (m, 1H), 7.62 (ddd, $J$ = 8.1, 6.9, 1.2 Hz, 1H), 7.09–7.04 (m, 1H), 6.95 (ddd, $J$ = 10.1, 8.7, 2.4 Hz, 1H). $^{13}$C{$^1$H} NMR (101 MHz, CDCl$_3$) $\delta$ ppm: 190.3, 163.2 (ddd, $J$ = 438.1, 256.3, 12.1 Hz), 149.7 (d, $J$ = 13.8 Hz), 138.8 (d, $J$ = 2.0 Hz), 132.9 (dd, $J$ = 10.4, 4.0 Hz), 132.2, 129.4 (d, $J$ = 11.5 Hz), 127.6, 126.6, 122.8 (dd, $J$ = 14.0, 3.7 Hz), 112.4 (dd, $J$ = 21.7, 3.6 Hz), 104.9 (t, $J$ = 25.6 Hz) $^{19}$F NMR (376 MHz, CDCl$_3$) $\delta$ ppm: −101.8, −104.5. HRMS m/z (ESI) calcd for C$_{16}$H$_9$F$_2$NO (M+H)$^+$ 270.0725, found 270.0723.

### 4.23. (2,4-Dichlorophenyl)(quinolin-3-yl)methanone (3aw)

Compound **3aw** was synthesized in accordance with the typical procedure. Purification using column chromatography on silica gel (PE:EA = 10:1) afforded product **3aw** (48.6 mg, 81%) as a white solid.

$^1$H NMR (400 MHz, CDCl$_3$) $\delta$ ppm: 9.30 (d, $J$ = 2.2 Hz, 1H), 8.45 (d, $J$ = 2.2 Hz, 1H), 8.16 (d, $J$ = 8.5 Hz, 1H), 7.87 (t, $J$ = 8.3 Hz, 2H), 7.61 (t, $J$ = 7.6 Hz, 1H), 7.52 (s, 1H), 7.42 (s, 2H). $^{13}$C{$^1$H} NMR (101 MHz, CDCl$_3$) $\delta$ ppm: 192.8, 149.9, 149.7, 139.4, 137.4, 135.9, 132.5, 130.5, 130.2, 129.5, 128.7, 127.7, 127.5, 126.6. The NMR spectra are consistent with the reported literature [31].

### 4.24. Benzo[d][1,3]dioxol-5-yl(quinolin-3-yl)methanone (3ax)

Compound **3ax** was synthesized in accordance with the typical procedure. Purification using column chromatography on silica gel (PE:EA = 10:1) afforded product **3ax** (43.6 mg, 79%) as a white solid.

$^1$H NMR (400 MHz, CDCl$_3$) $\delta$ ppm: 9.24 (d, $J$ = 2.2 Hz, 1H), 8.49 (d, $J$ = 2.2 Hz, 1H), 8.17 (d, $J$ = 8.5 Hz, 1H), 7.90 (d, $J$ = 8.1 Hz, 1H), 7.82 (t, $J$ = 7.7 Hz, 1H), 7.61 (t, $J$ = 7.5 Hz, 1H), 7.40 (d, $J$ = 1.8 Hz, 2H), 6.88 (d, $J$ = 8.5 Hz, 1H), 6.08 (s, 2H). $^{13}$C{$^1$H} NMR (101 MHz, CDCl$_3$) $\delta$ ppm: 193.0, 152.0, 150.1, 149.2, 148.3, 138.1, 131.6, 131.4, 130.6, 129.4, 129.0, 127.5, 127.0, 126.5, 109.6, 107.9, 102.0. The NMR spectra are consistent with the reported literature [22].

### 4.25. Naphthalen-2-yl(quinolin-3-yl)methanone (**3ay**)

Compound **3ay** was synthesized in accordance with the typical procedure. Purification using column chromatography on silica gel (PE:EA = 10:1) afforded product **3ay** (48.7 mg, 84%) as a white solid.

$^{1}$H NMR (400 MHz, CDCl$_3$) $\delta$ ppm: 9.38 (d, $J$ = 2.2 Hz, 1H), 8.61 (d, $J$ = 2.8 Hz, 1H), 8.31 (s, 1H), 8.22 (d, $J$ = 9.5 Hz, 1H), 7.99 (d, $J$ = 1.3 Hz, 2H), 7.95–7.82 (m, 4H), 7.68–7.54 (m, 3H). $^{13}$C{$^{1}$H} NMR (101 MHz, CDCl$_3$) $\delta$ ppm: 194.8, 150.3, 149.4, 138.8, 135.4, 134.2, 132.2, 132.1, 131.8, 130.3, 129.4, 129.1, 128.7, 128.7, 127.8, 127.6, 127.0, 126.6, 125.4. The NMR spectra are consistent with the reported literature [31].

### 4.26. Quinolin-3-yl(thiophen-2-yl)methanone (**3az**)

Compound **3az** was synthesized in accordance with the typical procedure. Purification using column chromatography on silica gel (PE:EA = 10:1) afforded product **3az** (38.6 mg, 80%) as a white solid.

$^{1}$H NMR (400 MHz, CDCl$_3$) $\delta$ ppm: 9.43 (d, $J$ = 2.2 Hz, 1H), 8.80 (d, $J$ = 2.2 Hz, 1H), 8.15 (d, $J$ = 8.4 Hz, 1H), 7.93 (d, $J$ = 8.1 Hz, 1H), 7.81 (t, $J$ = 7.1 Hz, 1H), 7.74 (s, 1H), 7.61 (t, $J$ = 7.5 Hz, 1H), 7.36 (d, $J$ = 3.6 Hz, 1H), 6.64 (d, $J$ = 3.7 Hz, 1H). $^{13}$C{$^{1}$H} NMR (101 MHz, CDCl$_3$) $\delta$ ppm: 180.3, 152.3, 149.7, 149.4, 147.4, 138.2, 131.8, 129.6, 129.4, 129.2, 127.5, 126.7, 120.7, 112.6. The NMR spectra are consistent with the reported literature [31].

### 4.27. Furan-2-yl(quinolin-3-yl)methanone (**3az′**)

Compound **3az′** was synthesized in accordance with the typical procedure. Purification using column chromatography on silica gel (PE:EA = 10:1) afforded product **3az′** (36.8 mg, 83%) as a white solid.

$^{1}$H NMR (400 MHz, CDCl$_3$) $\delta$ ppm: 9.33 (d, $J$ = 2.2 Hz, 1H), 8.63 (d, $J$ = 2.2 Hz, 1H), 8.17 (d, $J$ = 8.5 Hz, 1H), 7.93 (d, $J$ = 8.2 Hz, 1H), 7.83 (t, $J$ = 8.4 Hz, 1H), 7.78 (d, $J$ = 4.9 Hz, 1H), 7.71 (s, 1H), 7.63 (t, $J$ = 7.5 Hz, 1H), 7.20 (t, $J$ = 4.4 Hz, 1H). $^{13}$C{$^{1}$H} NMR (101 MHz, CDCl$_3$) $\delta$ ppm: 186.1, 149.5, 149.4, 143.1, 137.6, 135.0 (2C), 131.7, 130.6, 129.4, 129.0, 128.3, 127.6, 126.6. The NMR spectra are consistent with the reported literature [31].

### 4.28. (6-Methoxyquinolin-3-yl)(phenyl)methanone (**3ba**)

Compound **3ba** was synthesized in accordance with the typical procedure. Purification using column chromatography on silica gel (PE:EA = 10:1) afforded product **3ba** (46.1 mg, 83%) as a white solid.

$^{1}$H NMR (400 MHz, CDCl$_3$) $\delta$ ppm: 9.14 (d, $J$ = 2.1 Hz, 1H), 8.44 (d, $J$ = 2.2 Hz, 1H), 8.07 (d, $J$ = 9.8 Hz, 1H), 7.87–7.84 (m, 2H), 7.67–7.62 (m, 1H), 7.55–7.46 (m, 3H), 7.14 (d, $J$ = 2.8 Hz, 1H), 3.93 (s, 3H). $^{13}$C{$^{1}$H} NMR (101 MHz, CDCl$_3$) $\delta$ ppm: 195.1, 158.4, 147.9, 145.6, 137.4, 137.1, 133.0, 130.7, 130.3, 130.0, 128.6, 127.8, 124.8, 106.0, 55.6. The NMR spectra are consistent with the reported literature [31].

### 4.29. (6-Fluoroquinolin-3-yl)(phenyl)methanone (**3ca**)

Compound **3ca** was synthesized in accordance with the typical procedure. Purification using column chromatography on silica gel (PE:EA = 10:1) afforded product **3ca** (42.5 mg, 85%) as a white solid. $^{1}$H NMR (400 MHz, CDCl$_3$) $\delta$ ppm: 9.26 (d, $J$ = 2.1 Hz, 1H), 8.49 (d, $J$ = 2.2 Hz, 1H), 8.19 (dd, $J$ = 9.2, 5.2 Hz, 1H), 7.85 (dd, $J$ = 8.3, 1.3 Hz, 2H), 7.68–7.59 (m, 2H), 7.56–7.51 (m, 3H). $^{13}$C{$^{1}$H} NMR (101 MHz, CDCl$_3$) $\delta$ ppm: 194.6, 160.9 (d, $J$ = 250.3 Hz), 149.6 (d, $J$ = 2.8 Hz), 137.9 (d, $J$ = 5.6 Hz), 136.7, 133.2, 132.0 (d, $J$ = 9.2 Hz), 130.7, 130.0, 128.7, 127.3 (d, $J$ = 10.2 Hz), 122.0 (d, $J$ = 25.8 Hz), 112.0 (d, $J$ = 21.8 Hz). $^{19}$F NMR (376 MHz, CDCl$_3$) $\delta$ ppm: −111.2. The NMR spectra are consistent with the reported literature [31].

### 4.30. (6-Chloroquinolin-3-yl)(phenyl)methanone (**3da**)

Compound **3da** was synthesized in accordance with the typical procedure. Purification using column chromatography on silica gel (PE:EA = 10:1) afforded product **3da** (43.3 mg, 83%) as a white solid.

<sup>1</sup>H NMR (400 MHz, CDCl₃) δ ppm: 9.31 (d, *J* = 2.1 Hz, 1H), 8.45 (d, *J* = 2.1 Hz, 1H), 8.10–8.04 (m, 2H), 7.92–7.84 (m, 3H), 7.69–7.65 (m, 1H), 7.55 (t, *J* = 7.7 Hz, 2H). <sup>13</sup>C{<sup>1</sup>H} NMR (101 MHz, CDCl₃) δ ppm: 194.5, 150.4, 147.7, 137.7, 136.7, 133.4, 133.3, 132.7, 131.0, 130.8, 130.0, 128.7, 127.6, 127.2. The NMR spectra are consistent with the reported literature [31].

### 4.31. (6-Bromoquinolin-3-yl)(phenyl)methanone (3ea)

Compound **3ea** was synthesized in accordance with the typical procedure. Purification using column chromatography on silica gel (PE:EA = 10:1) afforded product **3ea** (54.2 mg, 87%) as a white solid.

<sup>1</sup>H NMR (400 MHz, CDCl₃) δ ppm: 9.31 (d, *J* = 2.1 Hz, 1H), 8.45 (d, *J* = 2.1 Hz, 1H), 8.10–8.04 (m, 2H), 7.91 (dd, *J* = 8.9, 2.2 Hz, 1H), 7.87–7.83 (m, 2H), 7.69–7.65 (m, 1H), 7.55 (t, *J* = 7.7 Hz, 2H). <sup>13</sup>C{<sup>1</sup>H} NMR (101 MHz, CDCl₃) δ ppm: 194.5, 150.6, 148.0, 137.6, 136.7, 135.2, 133.3, 131.1, 131.0, 130.7, 130.0, 128.7, 127.7, 121.5. The NMR spectra are consistent with the reported literature [31].

### 4.32. (7-Fluoroquinolin-3-yl)(phenyl)methanone (3fa)

Compound **3fa** was synthesized in accordance with the typical procedure. Purification using column chromatography on silica gel (PE:EA = 10:1) afforded product **3fa** (44.1 mg, 88%) as a white solid.

<sup>1</sup>H NMR (400 MHz, CDCl₃) δ ppm: 9.31 (d, *J* = 2.2 Hz, 1H), 8.55 (d, *J* = 2.4 Hz, 1H), 7.93 (dd, *J* = 9.0, 6.0 Hz, 1H), 7.86–7.79 (m, 3H), 7.68–7.63 (m, 1H), 7.54 (t, *J* = 7.7 Hz, 2H), 7.42 (td, *J* = 8.5, 2.5 Hz, 1H). <sup>13</sup>C{<sup>1</sup>H} NMR (101 MHz, CDCl₃) δ ppm: 194.5, 164.4 (d, *J* = 254.1 Hz), 151.4, 150.6 (d, *J* = 13.0 Hz), 138.6, 136.8, 133.1, 131.4 (d, *J* = 10.2 Hz), 123.0, 129.5, 128.7, 123.6, 118.3 (d, *J* = 25.6 Hz), 113.4 (d, *J* = 20.6 Hz) <sup>19</sup>F NMR (376 MHz, CDCl₃) δ ppm: −104.8. The NMR spectra are consistent with the reported literature [31].

### 4.33. (7-Chloroquinolin-3-yl)(phenyl)methanone (3ga)

Compound **3ga** was synthesized in accordance with the typical procedure. Purification using column chromatography on silica gel (PE:EA = 10:1) afforded product **3ga** (44.8 mg, 84%) as a white solid.

<sup>1</sup>H NMR (400 MHz, CDCl₃) δ ppm: 9.32 (d, *J* = 2.2 Hz, 1H), 8.54 (d, *J* = 2.2 Hz, 1H), 8.19 (d, *J* = 2.0 Hz, 1H), 7.88–7.84 (m, 3H), 7.69–7.64 (m, 1H), 7.61–7.52 (m, 3H). <sup>13</sup>C{<sup>1</sup>H} NMR (101 MHz, CDCl₃) δ ppm: 194.5, 151.3, 149.7, 138.5, 137.9, 136.7, 133.2, 130.3, 130.1, 130.0, 128.8, 128.7, 128.5, 125.0. The NMR spectra are consistent with the reported literature [31].

### 4.34. (7-Bromoquinolin-3-yl)(phenyl)methanone (3ha)

Compound **3ha** was synthesized in accordance with the typical procedure. Purification using column chromatography on silica gel (PE:EA = 10:1) afforded product **3ha** (52.1 mg, 83%) as a white solid.

<sup>1</sup>H NMR (400 MHz, CDCl₃) δ ppm: 9.31 (d, *J* = 2.1 Hz, 1H), 8.53 (dd, *J* = 2.1, 0.8 Hz, 1H), 8.38 (d, *J* = 2.0 Hz, 1H), 7.85 (dd, *J* = 8.3, 1.3 Hz, 2H), 7.79 (d, *J* = 8.6 Hz, 1H), 7.73 (dd, *J* = 8.7, 1.9 Hz, 1H), 7.69–7.65 (m, 1H), 7.57–7.53 (m, 2H). <sup>13</sup>C{<sup>1</sup>H} NMR (101 MHz, CDCl₃) δ ppm: 194.5, 151.3, 149.9, 138.6, 136.7, 133.2, 131.9, 131.3, 130.3, 130.2, 130.0, 128.7, 126.3, 125.2. The NMR spectra are consistent with the reported literature [31].

### 4.35. Phenyl(7-(trifluoromethyl)quinolin-3-yl)methanone (3ia)

Compound **3ia** was synthesized in accordance with the typical procedure. Purification using column chromatography on silica gel (PE:EA = 10:1) afforded product **3ia** (47.1 mg, 80%) as a white solid, m.p. 88–92 °C; <sup>1</sup>H NMR (400 MHz, CDCl₃) δ ppm: 9.39 (d, *J* = 2.2 Hz, 1H), 8.60 (d, *J* = 2.2 Hz, 1H), 8.50 (s, 1H), 8.06 (d, *J* = 8.2 Hz, 1H), 7.88–7.85 (m, 2H), 7.81 (dd, *J* = 8.5, 1.8 Hz, 1H), 7.68 (t, *J* = 7.5 Hz, 1H), 7.56 (t, *J* = 7.6 Hz, 2H). <sup>13</sup>C{<sup>1</sup>H} NMR (101 MHz, CDCl₃) δ ppm: 194.4, 151.5, 148.4, 138.2, 136.5, 133.4, 133.2 (q, *J* = 33.0 Hz), 131.7, 130.3, 130.0, 128.8, 128.1, 127.3 (q, *J* = 4.3 Hz), 123.6 (q, *J* = 271.5 Hz), 123.2 (d, *J* = 3.0 Hz). <sup>19</sup>F

NMR (376 MHz, CDCl$_3$) $\delta$ ppm: $-62.8$. HRMS *m/z* (ESI) calcd for C$_{17}$H$_{10}$F$_3$NO (M+H)$^+$ 302.0787, found 302.0779.

### 4.36. Methyl 3-benzoylquinoline-7-carboxylate (*3ja*)

Compound **3ja** was synthesized in accordance with the typical procedure. Purification using column chromatography on silica gel (PE:EA = 10:1) afforded product **3ja** (43.6 mg, 75%) as a white solid, m.p. 142–145 °C; $^1$H NMR (400 MHz, CDCl$_3$) $\delta$ ppm: 9.36 (d, *J* = 2.2 Hz, 1H), 8.87 (s, 1H), 8.56 (d, *J* = 3.1 Hz, 1H), 8.21 (dd, *J* = 8.5, 1.7 Hz, 1H), 7.97 (d, *J* = 8.9 Hz, 1H), 7.88–7.85 (m, 2H), 7.69–7.64 (m, 1H), 7.57–7.52 (m, 2H), 4.02 (s, 3H).$^{13}$C{$^1$H} NMR (101 MHz, CDCl$_3$) $\delta$ ppm: 194.5, 166.3, 151.1, 148.8, 138.2, 136.6, 133.3, 132.8, 131.8, 131.4, 130.0, 129.3, 129.0, 128.7, 127.0, 52.7. HRMS *m/z* (ESI) calcd for C$_{18}$H$_{13}$NO$_3$ (M+H)$^+$ 292.0968, found 292.0961.

### 4.37. (6,8-Dichloroquinolin-3-yl)(phenyl)methanone (*3ka*)

Compound **3ka** was synthesized in accordance with the typical procedure. Purification using column chromatography on silica gel (PE:EA = 10:1) afforded product **3ka** (46.5 mg, 78%) as a white solid.

$^1$H NMR (400 MHz, CDCl$_3$) $\delta$ ppm: 9.37 (d, *J* = 2.1 Hz, 1H), 8.49 (d, *J* = 2.1 Hz, 1H), 7.94 (d, *J* = 2.2 Hz, 1H), 7.87–7.83 (m, 3H), 7.68 (t, *J* = 7.4 Hz, 1H), 7.55 (t, *J* = 7.8 Hz, 2H). $^{13}$C{$^1$H} NMR (101 MHz, CDCl$_3$) $\delta$ ppm: 194.0, 150.9, 144.1, 138.1, 137.8, 136.4, 135.0, 133.6, 132.9, 132.2, 131.7, 130.1, 128.8, 128.3, 126.7, 126.5. The NMR spectra are consistent with the reported literature [16].

### 4.38. (6,7-Dimethoxyquinolin-3-yl)(phenyl)methanone (*3la*)

Compound **3la** was synthesized in accordance with the typical procedure. Purification using column chromatography on silica gel (PE:EA = 10:1) afforded product **3la** (46.8 mg, 80%) as a white solid. $^1$H NMR (400 MHz, CDCl$_3$) $\delta$ ppm: 9.11 (d, *J* = 2.1 Hz, 1H), 8.41 (d, *J* = 2.9 Hz, 1H), 7.85–7.82 (m, 2H), 7.64–7.59 (m, 1H), 7.54–7.47 (m, 3H), 7.10 (s, 1H), 4.06 (s, 3H), 4.00 (s, 3H). $^{13}$C{$^1$H} NMR (101 MHz, CDCl$_3$) $\delta$ ppm: 195.1, 154.4, 150.4, 148.5, 147.0, 137.3, 136.8, 132.7, 129.9, 128.5, 122.3, 107.8, 106.0, 56.3, 56.1. The NMR spectra are consistent with the reported literature [31].

### 4.39. [1,3]Dioxolo [4,5-g]quinolin-7-yl(phenyl)methanone (*3ma*)

Compound **3ma** was synthesized in accordance with the typical procedure. Purification using column chromatography on silica gel (PE:EA = 10:1) afforded product **3ma** (45.1 mg, 81%) as a white solid. $^1$H NMR (400 MHz, CDCl$_3$) $\delta$ ppm: 9.08 (d, *J* = 2.2 Hz, 1H), 8.37 (d, *J* = 2.1 Hz, 1H), 7.85–7.81 (m, 2H), 7.63 (t, *J* = 7.4 Hz, 1H), 7.52 (t, *J* = 7.6 Hz, 2H), 7.43 (s, 1H), 7.11 (s, 1H), 6.15 (s, 2H). $^{13}$C{$^1$H} NMR (101 MHz, CDCl$_3$) $\delta$ ppm: 194.9, 152.7, 148.6, 148.5, 148.4, 137.4, 137.3, 137.2, 132.8, 128.6, 128.5, 123.8, 105.8, 103.6, 102.2. The NMR spectra are consistent with the reported literature [31].

### 4.40. (6-Chloro-4-phenylquinolin-3-yl)(phenyl)methanone (*3na*)

Compound **3na** was synthesized in accordance with the typical procedure. Purification using column chromatography on silica gel (PE:EA = 10:1) afforded product **3na** (57.0 mg, 84%) as a white solid. $^1$H NMR (400 MHz, CDCl$_3$) $\delta$ ppm: 8.96 (s, 1H), 8.17 (d, *J* = 9.7 Hz, 1H), 7.73 (dd, *J* = 6.9, 2.4 Hz, 2H), 7.59 (dd, *J* = 8.3, 1.4 Hz, 2H), 7.44 (t, *J* = 7.4 Hz, 1H), 7.34–7.27 (m, 5H), 7.26–7.23 (m, 2H). $^{13}$C{$^1$H} NMR (101 MHz, CDCl$_3$) $\delta$ ppm: 196.4, 148.6, 147.1, 146.1, 137.0, 134.2, 133.6, 133.4, 132.5, 131.4, 129.9, 129.7, 128.8, 128.4, 128.3, 127.2, 125.4. The NMR spectra are consistent with the reported literature [61].

**Supplementary Materials:** The following supporting information can be downloaded at https://www.mdpi.com/article/10.3390/catal13040778/s1.

**Author Contributions:** Methodology, K.-L.Z.; writing—review and editing, K.-L.Z. and J.-C.Y.; review and editing, Q.G.; supervision, L.-H.Z.; project administration, L.-H.Z. All authors have read and agreed to the published version of the manuscript.

**Funding:** The financial support from the Open Research Fund of School of Chemistry and Chemical Engineering, Henan Normal University (2022B01) is greatly appreciated.

**Data Availability Statement:** Data supporting reported results can be found in Supplementary Materials.

**Conflicts of Interest:** The authors declare no conflict of interest.

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
