# Peer review of "Transition Metal-Free Synthesis of 3-Acylquinolines through Formal [4+2] Annulation of Anthranils and Enaminones"

_catalysts, doi:10.3390/catal13040778_

Round 1

Reviewer 1 Report

1)      It is recommended to cite appropriate references and related info on the synthesis of 3-substituted quinolines by using a different catalyst.

2)      How did the authors confirm the intermediate ‘D’ is formed in Scheme 5?

3)      What is the role of additive, and how the additive played a role in the reported reaction to perform higher yields?

4)      3-Substituted quinolines are obtained through aza-Michael addition, hence, authors should represent the aza-Michael addition process in scheme 5.

Author Response

Response to Reviewer 1 Comments

Point 1:   It is recommended to cite appropriate references and related info on the synthesis of 3-substituted quinolines by using a different catalyst.

Response 1: Thanks for the reviewer’s comments. We added one paragraph to describe the synthesis of 3-arcylquinolines and cited some references.

Point 2:    How did the authors confirm the intermediate ‘D’ is formed in Scheme 5?

Response 2: Thanks for the reviewer’s comments. This might be a Michael addition process.

Point 3:    What is the role of additive, and how the additive played a role in the reported reaction to perform higher yields?

Response 3: Thanks for the reviewer’s comments. We explained the possible reason in the text.

Point 4:     3-Substituted quinolines are obtained through aza-Michael addition, hence, authors should represent the aza-Michael addition process in scheme 5.

Response 4: Thanks for the reviewer’s comments. We revised the mechanism.

Reviewer 2 Report

The manuscript by Liang-Hua Zou and coworkers describes a new approach to synthesize 3-substituted quinolines obtained through [4+2] annulation of anthranils and enaminones. The protocol seems to be very fruitful, providing desired products in moderate to excellent yields. The research is well conducted, and the compounds obtained are well characterized by NMR and HRMS analysis. However, this reviewer feels that this manuscript needs some minor revision, addressed in the following comments:

1. Additional reference on the synthesis of quinoline by metal-catalyzed reaction of anthanyls with enaminone is suggested: Org. Biomol. Chem., 2017, 15, 7387-7395.

2. The optimization of the reaction conditions describes the use of ethanol as solvent at 110 °C or 120 °C. I suggest to the authors revise this point or better explain the use of a high temperature in the presence of a low boiling point solvent.

3. Page 2, line 75: “naphtyl group” instead of “napht-group”.

4. Page 4, Scheme 3: Please check the information above the arrow in scheme 3.

4. Page 5, Scheme 5: Did the authors attempted to detect evidence of the C and D intermediates in the reaction mixture, as their formation is suggested? According to Table 1, row 10, the use of the NaI reagent is a very important factor for obtaining quinolines in high yields. However, its role during the reaction is neither discussed nor presented in the mechanism proposal.

5. In Experimental section please state which compound was used as standard to determine the 19F NMR chemical shifts.

6. Authors should thoroughly check the values of spin-spin coupling constants through all the Experimental section taking into account the fact that the spin-spin coupling constants of the protons, which are coupled to each other, should be the same.

Author Response

Response to Reviewer 2 Comments

The manuscript by Liang-Hua Zou and coworkers describes a new approach to synthesize 3-substituted quinolines obtained through [4+2] annulation of anthranils and enaminones. The protocol seems to be very fruitful, providing desired products in moderate to excellent yields. The research is well conducted, and the compounds obtained are well characterized by NMR and HRMS analysis. However, this reviewer feels that this manuscript needs some minor revision, addressed in the following comments:

Point 1:   Additional reference on the synthesis of quinoline by metal-catalyzed reaction of anthanyls with enaminone is suggested: Org. Biomol. Chem., 2017, 15, 7387-7395.

Response 1: Thanks for the reviewer’s comments. We added this reference.

Point 2:   The optimization of the reaction conditions describes the use of ethanol as solvent at 110 °C or 120 °C. I suggest to the authors revise this point or better explain the use of a high temperature in the presence of a low boiling point solvent.

Response 2: Thanks for the reviewer’s comments. We think that this is normal in synthetic reactions. Maybe the quick reflux under such conditions is better for the reaction.

Point 3:  Page 2, line 75: “naphtyl group” instead of “napht-group”.

Response 3: Thanks for the reviewer’s comments. We have revised it.

Point 4: Page 4, Scheme 3: Please check the information above the arrow in scheme 3.

Response 4: Thanks for the reviewer’s comments. We have revised it.

Point 5:  Page 5, Scheme 5: Did the authors attempted to detect evidence of the C and D intermediates in the reaction mixture, as their formation is suggested? According to Table 1, row 10, the use of the NaI reagent is a very important factor for obtaining quinolines in high yields. However, its role during the reaction is neither discussed nor presented in the mechanism proposal.

Response 5: Thanks for the reviewer’s comments. We revised the mechanism and added one sentence to explain the role of NaI.

Point 6:   In Experimental section please state which compound was used as standard to determine the 19F NMR chemical shifts.

Response 6: Thanks for the reviewer’s comments. We have added.

Point 7:  Authors should thoroughly check the values of spin-spin coupling constants through all the Experimental section taking into account the fact that the spin-spin coupling constants of the protons, which are coupled to each other, should be the same.

Response 7: Thanks for the reviewer’s comments. We have checked and revised.

Reviewer 3 Report

Zou group reports a transition metal-free protocol for the synthesis of 3-substituted quinolines through aza-Michael addition and intramolecular annulation of enaminones with anthranils. In the process, both methanesulfonic acid (MSA) and NaI play an important role in the reaction. This protocol features easy operation, high yields, broad substrate scope and excellent efficiency. This reviewer recommends acceptance of this manuscript after minor revision.

1. Some format errors exist: Line 65, page 2, 3ag-3ai; Line 265, page 8, 11H NMR; Line 491, page 11, 3da; Line 114, page 4, indicate should be indicated.

2. Line 86, page 3, product should be products.

3. Line 69, page 2, significant should be significantly.

4. Line 111, page 4, a ratio should be the ratio.

5. One example in Table 1 should be shown: how is the result when no additive was added in the reaction.

Author Response

Response to Reviewer 3 Comments

Zou group reports a transition metal-free protocol for the synthesis of 3-substituted quinolines through aza-Michael addition and intramolecular annulation of enaminones with anthranils. In the process, both methanesulfonic acid (MSA) and NaI play an important role in the reaction. This protocol features easy operation, high yields, broad substrate scope and excellent efficiency. This reviewer recommends acceptance of this manuscript after minor revision.

Point 1: Some format errors exist: Line 65, page 2, 3ag-3ai; Line 265, page 8, 11H NMR; Line 491, page 11, 3da; Line 114, page 4, indicate should be indicated.

Response 1: Thanks for the reviewer’s comments. We have checked and revised.

Point 2: Line 86, page 3, product should be products.

Response 2: Thanks for the reviewer’s comments. We have revised.

Point 3: Line 69, page 2, significant should be significantly.

Response 3: Thanks for the reviewer’s comments. We have revised.

Point 4: Line 111, page 4, a ratio should be the ratio.

Response 4: Thanks for the reviewer’s comments. We have revised.

  1. One example in Table 1 should be shown: how is the result when no additive was added in the reaction.

Response 5: Thanks for the reviewer’s comments. We added one entry (as entry 8) in Table 1.

Reviewer 4 Report

In this manuscript, the authors developed a cascade annulation of anthranils and enaminones, and a series of 3-substituted quinolines were obtained with good yields Furthermore, some control experiments were conducted to elucidate the reaction mechanism. This work provides a useful protocol for constructing quinolines under transition metal-free conditions. Due to the importance of quinolines in natural products and bioactive molecules, developing efficient methods for constructing quinolines is highly valuable. In addition, the Supporting Information is in a high quality. Therefore, this work is recommended for publication in Catalysts after minor revisions.

1. The C-F coupling should be provided for compounds, such as 3ag, 3ap, 3av.

2. Can the aryl groups enaminones 2 be replaced by alkyl groups? If the authors have tried and have some information, it is suggested that they briefly mention it in the main text or SI, because the readers might be interested in it even if such substrates cannot work for this reaction.

3. Some recent examples of transition metal-free synthesis of quinolines should be cited. For instance, Chin. J. Chem., 2022, 40, 365; Chin. J. Chem., 2022, 40, 71; Angew. Chem., Int. Ed. 2022, 61, e202112226.

4. It is suggested that the authors change “[4+2] annulation” into “formal [4+2] annulation” throughout the manuscript.

Author Response

Response to Reviewer 4 Comments

In this manuscript, the authors developed a cascade annulation of anthranils and enaminones, and a series of 3-substituted quinolines were obtained with good yields Furthermore, some control experiments were conducted to elucidate the reaction mechanism. This work provides a useful protocol for constructing quinolines under transition metal-free conditions. Due to the importance of quinolines in natural products and bioactive molecules, developing efficient methods for constructing quinolines is highly valuable. In addition, the Supporting Information is in a high quality. Therefore, this work is recommended for publication in Catalysts after minor revisions.

Point 1: The C-F coupling should be provided for compounds, such as 3ag3ap3av.

Response 1: Thanks for the reviewer’s comments. The C-F coupling have been provided.

Point 2: Can the aryl groups enaminones 2 be replaced by alkyl groups? If the authors have tried and have some information, it is suggested that they briefly mention it in the main text or SI, because the readers might be interested in it even if such substrates cannot work for this reaction.

Response 2: Thanks for the reviewer’s comments. We tried the results were not good.

Point 3: Some recent examples of transition metal-free synthesis of quinolines should be cited. For instance, Chin. J. Chem., 2022, 40, 365; Chin. J. Chem., 2022, 40, 71; Angew. Chem., Int. Ed. 2022, 61, e202112226.

Response 3: Thanks for the reviewer’s comments. These references have been added.

Point 4:  It is suggested that the authors change “[4+2] annulation” into “formal [4+2] annulation” throughout the manuscript.

Response 4: Thanks for the reviewer’s comments. We have revised.

Reviewer 5 Report

Zou and coworkers reported a solid protocol for the synthesis of 3-substituted quinolines through metal-free annulation process between anthranils and enaminones. Quinolines are highly requested chemical motifs for many scopes. In virtue of that, in my opinion, this work may be of high interest to the organic chemist audience. The introduction contains an exhaustive number of citations and is well organized, as well as the result section, where different cohorts of substrates were subjected to the transformation without significant alteration of reactivity. The conclusions are appropriate. I think this work has what is needed for a publication in Catalysis, however, a few points have to be addressed and answered prior to publication. Please find the details below. 

-        Page 2, line 51; page 3, line 68: “did not” instead

-        During the optimization, iodine salts revealed to be the best choice for a better yield. Why is that? Iodine is clearly taking an active role into the mechanism, please highlight it. 

-        Have you tried other acids than MSA? If yes, please provide a table regarding this point (in Supporting Information is fine)

-        Table 1 entry 6. The use of THF gave comparable yield (29% vs 31% reported for entry 1). What happens if you add MSA/NaI and THF instead of EtOH? Maybe you can low down the reaction temperature. 

-        There is nowadays a trend oriented to bio-sourced solvents for a greener approach to organic transformations: have you tried Me-THF for example?

-        For the optimized conditions, a protic acid is also used. How are you sure that intermediate A can be obtained upon protonation by MSA? In solution there is also NaI. How can be excluded that you get HI before, which protonates the carbonyl afterward instead? Please make a comment on that and modify accordingly the mechanistic explanation.

-        Proposed mechanism (scheme 5): please highlight the role of NaI.

Supporting informations:

-        1H-NMR spectra of 3aa, 3ac, 3ad, 3ae, 3ag, 3ah, 3ai, 3aj, 3ak, 3al, 3am, 3an, 3ao, 3ap, 3aq, 3ar, 3as, 3at, 3au, 3av, 3aw, 3ay (requires also a phase correction), 3az, 3az’, 3ba, 3da, 3ea, 3fa, 3ga, 3ha, 3ia, 3ja, 3ka, 3ma, 3na contain too many impurities in the aliphatic region and cannot be accepted. Please provide new spectra. 

-        13C-NMR spectra of 3ae, 3ao, 3ap, 3aq, 3au, 3av, 3aw, 3ay, 3da, 3ga, 3ia contain too many impurities in the aliphatic region and cannot be accepted. Please provide new spectra.

Author Response

Response to Reviewer 5 Comments

Zou and coworkers reported a solid protocol for the synthesis of 3-substituted quinolines through metal-free annulation process between anthranils and enaminones. Quinolines are highly requested chemical motifs for many scopes. In virtue of that, in my opinion, this work may be of high interest to the organic chemist audience. The introduction contains an exhaustive number of citations and is well organized, as well as the result section, where different cohorts of substrates were subjected to the transformation without significant alteration of reactivity. The conclusions are appropriate. I think this work has what is needed for a publication in Catalysis, however, a few points have to be addressed and answered prior to publication. Please find the details below. 

Point 1: Page 2, line 51; page 3, line 68: “did not” instead

Response 1: Thanks for the reviewer’s comments. We have revised.

Point 2:    During the optimization, iodine salts revealed to be the best choice for a better yield. Why is that? Iodine is clearly taking an active role into the mechanism, please highlight it. 

Response 2: Thanks for the reviewer’s comments. We revised the machanism and added one sentence to explain the role of NaI.

Point 3:    Have you tried other acids than MSA? If yes, please provide a table regarding this point (in Supporting Information is fine)

Response 3: Thanks for the reviewer’s comments. We added more details in SI.

Point 4:    Table 1 entry 6. The use of THF gave comparable yield (29% vs 31% reported for entry 1). What happens if you add MSA/NaI and THF instead of EtOH? Maybe you can low down the reaction temperature. 

Response 4: Thanks for the reviewer’s comments. We added one entry (as entry 10) in Table 1.

Point 5:     There is nowadays a trend oriented to bio-sourced solvents for a greener approach to organic transformations: have you tried Me-THF for example?

Response 5: Thanks for the reviewer’s comments. We did not try this solvent.

Point 6:    For the optimized conditions, a protic acid is also used. How are you sure that intermediate A can be obtained upon protonation by MSA? In solution there is also NaI. How can be excluded that you get HI before, which protonates the carbonyl afterward instead? Please make a comment on that and modify accordingly the mechanistic explanation.

Response 6: Thanks for the reviewer’s comments. We tried some weak acid like AcOH and hexanoic acid, but it did not work. Therefore, HI might not be the key factor. According to some reference, TsOH or MSA can activate ketones.

Point 7:     Proposed mechanism (scheme 5): please highlight the role of NaI.

 Response 7: Thanks for the reviewer’s comments. We added one sentence to highlight the role of NaI.  

Point 8: Supporting informations:

-        1H-NMR spectra of 3aa, 3ac, 3ad, 3ae, 3ag, 3ah, 3ai, 3aj, 3ak, 3al, 3am, 3an, 3ao, 3ap, 3aq, 3ar, 3as, 3at, 3au, 3av, 3aw, 3ay (requires also a phase correction), 3az, 3az’, 3ba, 3da, 3ea, 3fa, 3ga, 3ha, 3ia, 3ja, 3ka, 3ma, 3na contain too many impurities in the aliphatic region and cannot be accepted. Please provide new spectra. 

-        13C-NMR spectra of 3ae, 3ao, 3ap, 3aq, 3au, 3av, 3aw, 3ay, 3da, 3ga, 3ia contain too many impurities in the aliphatic region and cannot be accepted. Please provide new spectra.

Response 8: Thanks for the reviewer’s comments. We have tried our best, but a few inpurities from petroleum ether was difficult to remove.